# Efficacy of Exercise-Based Rehabilitation Programs for Improving Muscle Function and Size in People with Hip Osteoarthritis: A Systematic Review with Meta-Analysis

**DOI:** 10.3390/biology10121251

**Published:** 2021-11-30

**Authors:** Zachary P. J. Rostron, Rodney A. Green, Michael Kingsley, Anita Zacharias

**Affiliations:** 1Department of Pharmacy and Biomedical Sciences, College of Science, Health and Engineering, La Trobe University, Bendigo, VIC 3550, Australia; rod.green@latrobe.edu.au (R.A.G.); a.zacharias@latrobe.edu.au (A.Z.); 2Department of Exercise Sciences, Faculty of Science, University of Auckland, Auckland 1023, New Zealand; michael.kingsley@aukland.ac.nz; 3Holsworth Research Initiative, College of Science, Health and Engineering, La Trobe University, Bendigo, VIC 3550, Australia

**Keywords:** lower limb, arthritis, exercise intensity, muscle volume, muscle strength

## Abstract

**Simple Summary:**

Hip osteoarthritis (OA) is characterised by increased pain and functional disability. People with hip OA also have reduced muscle size and weakness, especially in the muscles surrounding the hip. Current exercise-based rehabilitation programs aim to reduce clinical symptoms and improve physical function. High-intensity resistance exercises can improve muscles strength in people with knee OA; however, the efficacy of these exercises to improve muscle function and size in people with hip OA has received less attention. Therefore, the aim of this review was to identify whether current rehabilitation programs involving high-intensity exercises could improve muscle size and function in people with hip OA. The findings support increased hip abduction strength, favouring a high-intensity resistance intervention when compared to a control group in people with hip OA. However, no differences were observed in hip or knee muscle function or size when comparing high-intensity resistance to low-intensity resistance interventions. This review also highlighted the dearth of studies that evaluate muscle outcomes following current rehabilitation programs in people with hip OA. Therefore, future studies should include measurements of muscle function and size when evaluating the effects of current rehabilitation programs in people with hip OA.

**Abstract:**

Objective: To determine the effect of exercise-based rehabilitation programs on hip and knee muscle function and size in people with hip osteoarthritis. Methods: Seven databases were systematically searched in order to identify studies that assessed muscle function (strength or power) and size in people with hip osteoarthritis after exercise-based rehabilitation programs. Studies were screened for eligibility and assessed for quality of evidence using the GRADE approach. Data were pooled, and meta-analyses was completed on 7 of the 11 included studies. Results: Six studies reported hip and/or knee function outcomes, and two reported muscle volumes that could be included in meta-analyses. Meta-analyses were conducted for four strength measures (hip abduction, hip extension, hip flexion, and knee extension) and muscle size (quadriceps femoris volume). For hip abduction, there was a low certainty of evidence with a small important effect (effect size = 0.28, 95% CI = 0.01, 0.54) favouring high-intensity resistance interventions compared to control. There were no other comparisons or overall meta-analyses that identified benefits for hip or knee muscle function or size. Conclusion: High-intensity resistance programs may increase hip abduction strength slightly when compared with a control group. No differences were identified in muscle function or size when comparing a high versus a low intensity group. It is unclear whether strength improvements identified in this review are associated with hypertrophy or other neuromuscular factors.

## 1. Introduction

Osteoarthritis (OA) typically results in the progressive degeneration of articular cartilage [1] and is commonly observed in weight-bearing joints such as the hips or knees [2]. This chronic condition is prevalent with 2.2 million Australians diagnosed with OA in 2017–2018 [3]. The pain and physical dysfunction associated with OA can negatively impact quality of life [4]. Risk factors for OA include an increase in age, obesity, injury or trauma, and repetitive loading [5]. Osteoarthritis is also associated with muscle atrophy [6] and weakness [7,8], leading to functional disability [9]. Changes observed in individuals with hip OA include a reduction in gluteal muscle size reported both as cross-sectional area [10] and volume [6,11], and decreased hip muscle strength [12]. In people with hip OA, pain, muscle weakness, and changes in gluteal muscle activity can lead to an altered gait along with a reduced capacity to complete functional tasks (e.g., walking, climbing stairs) [9,13,14].

People with hip OA are commonly prescribed exercise-based rehabilitation programs that focus on reducing pain and improving physical function [15,16,17], as these are typical presenting clinical symptoms [16]. Exercise-based rehabilitation programs that include strength exercises, flexibility, and aerobic components have previously been identified to reduce pain, increase physical function, and improve overall quality of life in people with hip OA [18,19]. However, there is less evidence on expected exercise-based outcomes such as muscle strength (function) and hypertrophy following rehabilitation interventions in people with hip OA [20]. Exercise programs incorporating resistance-based exercises have previously been associated with muscle hypertrophy in a healthy population [21]. Manipulating factors such as frequency, intensity, time, and type of exercise in rehabilitation programs has the potential to influence muscle strength and hypertrophy [22]. Therefore, interventions including resistance exercise programs have the potential to improve functional outcomes in people with hip OA.

A previous systematic review and meta-analysis that included participants with hip and/or knee OA identified improved knee muscle strength in people with knee OA after completing high-intensity resistance exercise when compared to a control group [20]. Only one study, published in 1993, was identified in the aforementioned review that reported muscle strength outcomes for people with hip OA comparing two exercise interventions [23]. The authors of this study reported no additional strength improvements from adding hydrotherapy to a home-based exercise program; unfortunately, this study did not include a matched control group [23]. There has been an increased focus in recent years on exercise-based rehabilitations programs for people with hip OA [24] and a better understanding of gluteal muscle changes associated with hip OA [6,7,9,13]. Given the improvements in knee muscle strength associated with high-intensity exercise in individuals with knee OA identified by meta-analysis [20], it is possible that high-intensity exercise programs might produce similar positive outcomes for hip muscles in people with hip OA.

Therefore, the aim of this review was to determine whether high-intensity exercise improves muscle structure and function in people with hip OA by synthesising results from recent literature.

## 2. Materials and Methods

### 2.1. Search Strategy with Study Identification

Literature searches were systematically completed using seven databases (AUSPORT, CINAHL, COCHRANE, Embase, MEDLINE, PEDro, and SPORTDiscus) from February 2013 to October 2021. This search supplemented a previous search from the earliest possible date to February 2013 by the same team that found one study [23] meeting the inclusion criteria of the current search, and this study was also included in the current review. A keyword search was completed using three main concepts: “population”, “intervention”, and “body region” that were combined using the “AND” Boolean operator (Table 1). All keywords within each concept were combined using the “OR” operator (Table 1). The total search yield was then imported to Endnote X9 (Clarivate Analytic, Philadelphia, PA, USA, 2012) for removal of duplicates and screening against inclusion criteria (Table 2).

Titles and abstracts were screened independently by two reviewers. Subsequently, full-text articles were also independently screened by two reviewers to identify studies for inclusion and data extraction. Where differences of opinion existed regarding inclusion, consensus was reached following discussion. Citation tracking was completed using Google Scholar, and reference checking was completed for all included studies.

### 2.2. Study Selection

#### 2.2.1. Population

All included studies were restricted to human participants with hip OA. Studies that included patients with various other forms of arthritis (e.g., juvenile idiopathic arthritis) were excluded due to possible muscle dysfunction associated with these conditions [25].

#### 2.2.2. Interventions

Exercise-based interventions were required to be of at least six weeks duration to allow for muscle hypertrophy, rather than strength improvements seen due to neuromuscular factors (e.g., motor learning via initial muscle activation) in shorter interventions [26,27]. Exercise-based interventions were classified as high-intensity, low-intensity, aerobic, or multimodal on the basis of standard criteria [22,28] used in the previous review (Table 3) [20]. One study [29] reported their interventions as high- and low-velocity resistance training. Because these interventions did not fit the existing criteria, they were classified on the basis of the velocity of movement as high-intensity resistance and low-intensity resistance for this review [30].

#### 2.2.3. Comparisons

Exercise intervention groups were contrasted with a comparison group that could be either a control group or an alternative form of exercise that was different to the intervention group (e.g., high-intensity resistance versus low-intensity resistance).

#### 2.2.4. Outcomes

Muscle function (strength or power) and muscle size were the outcomes of interest and were assessed at various follow-up time points including short-term (ST: 6–13 weeks), intermediate-term (IT: 13–24 weeks), and long-term (LT: >24 weeks) [20]. Studies were included if they reported muscle strength (e.g., Nm/kg, N) and/or muscle size (e.g., cross-sectional area in cm^2^, muscle thickness in cm) of muscles acting across the hip or knee joint. Muscle power (i.e., W) was recorded for the purpose of muscle function only when strength data were not available. Various forms of muscle function testing (e.g., hand-held dynamometry, fixed dynamometer) and muscle size measures (e.g., MRI, ultrasound) were reported in the included studies. Studies that reported only on pain and functional outcomes were excluded from this review.

#### 2.2.5. Study Design

Included studies required participants to be randomised to one of at least two groups, written in the English language with muscle function and/or muscle size data and published in peer reviewed journals. Studies using non-randomised designs, reviews, and conference presentations were not included.

### 2.3. Data Extraction

Using a standard spread sheet (available upon request), one reviewer extracted data independently, and it was verified by a second reviewer. Extracted data included participant demographic characteristics (i.e., age and population characteristics), details of intervention programs (i.e., frequency, intensity, time, type), follow-up time point, and details of outcome measures of interest. Post-intervention data (i.e., mean ± SD) were extracted where possible, and where data were not reported, authors were contacted to request these data.

### 2.4. Data Analysis

Post-intervention mean ± SD were entered into Review Manager (RevMan Version 5.3; The Nordic Cochrane Centre, The Cochrane Collaboration, 2014) by one reviewer and verified by a second reviewer.

To compare different types of interventions across studies and determine their effectiveness on muscle function or size on patients with hip OA, studies were grouped according to intervention type and post-intervention follow-up time point, as described in the previous review [20]. Due to low study numbers, data were pooled across follow-up time points for meta-analysis using a random effects model. Where multiple follow-up time points were included in a study, only the ST was included in the meta-analysis in order to reduce heterogeneity in follow-up time points. High-intensity resistance programs were compared to all other interventions (control or other exercise interventions) because high-intensity resistance programs were shown to be most effective in improving muscle strength in people with knee OA in the previously published review [20].

On the basis of standard Cochrane guidelines [31], we used the standardised mean difference (SMD) with 95% CI to calculate the effect size for each outcome individually (e.g., hip abduction strength, knee extension strength). Reported effect sizes were classified as small (>0.2), medium (>0.5), or large (>0.8) [32].

### 2.5. Quality Assessment

The updated Cochranes risk of bias tool (RoB 2.0) was used to rate the methodological quality of all included studies [33]. Three judgment items were considered: low, unclear, and high risk of bias, across five RoB domains (bias arising from the randomisation process, any deviations from the intended interventions, missing outcome data, bias in measurement of outcomes, and selection of reported results) [33].

The Grading of Recommendation, Assessment, Development and Evaluation (GRADE) approach was used independently by one reviewer and verified by a second to evaluate the quality of the body of evidence for included outcomes within the meta-analysis [34]. The quality of evidence was rated as high, medium, low, and very low, according to five domains: risk of bias assessment, inconsistency, indirectness, and publication bias. Conclusions were presented according to formulated statements in previously published guidelines [35].

## 3. Results

### 3.1. Yield

Of the 4603 studies that were identified, 21 full-text studies were screened against the inclusion criteria (Figure 1). This resulted in a total of 11 included studies [19,23,29,36,37,38,39,40,41,42,43] (including one from the previous review) within the final yield (Table 4).

Studies mainly reported on functional outcomes for hip abductors, flexors, extensors and rotators, and knee extensors and flexors (Table 4). Exercise interventions were classified as high-intensity resistance (eight studies), low-intensity resistance (five studies), aerobic (one study), and multimodal (one study). Intervention durations ranged from 6 to 16 weeks for all studies, with follow-up time points reported for ST (ten studies), IT (three studies), and LT (two studies). Only two studies reported on muscle size [29,43].

Of the 11 included studies, 2 [29,38] failed to conceal allocation, 2 [37,38] did not conduct an intention to treat analysis, 1 [29] failed to blind the outcome assessor, and 7 [19,23,29,36,38,39,40] presented bias in selection of the reported results (Figure 2). However, there was a low risk of bias for the methodological quality across all included studies (Figure 3).

The certainty of quality of evidence in the meta-analyses ranged from very low to moderate for all comparisons, with one outcome measure demonstrating a moderate certainty of evidence (hip extension; Appendix A
Table A1) and three outcomes demonstrating low certainty (hip abduction, hip flexion, knee extension; Table A2, Table A3 and Table A4) when a high-intensity resistance was compared to a control. A low certainty of evidence was identified for three outcomes (hip abduction, hip flexion, and knee extension; Table A2, Table A3 and Table A4), and a very low certainty of evidence for one outcome (quadriceps femoris muscle size; Table A5) when comparing high-intensity resistance to low-intensity resistance.

### 3.2. Results of Meta-Analysis

Only 7 of the 11 included studies had sufficient data to be included within a meta-analysis [29,36,39,40,41,42,43].

#### High-Intensity Resistance Exercise vs. Comparison (i.e., Low-Intensity Resistance Exercise and/or Control)

Meta-analysis resulted in a low certainty of evidence with a small important effect (effect size = 0.28, 95% CI = 0.01, 0.54, P = 0.04, n = 2, N = 222, I^2^ = 0; Figure 4) favouring high-intensity resistance for hip abduction function when compared to a control group. However, a meta-analysis comparing high-intensity resistance to a low-intensity resistance group identified no differences between groups for hip abduction strength.

Although not significant, a similar pattern for hip flexion strength was observed following meta-analysis, with a larger effect size when comparing high-intensity resistance to a control group (effect size = 0.18, 95% CI = −0.17, 0.54, P = 0.31; Figure 4) than when comparing a high-intensity resistance to a low-intensity resistance (effect size = −0.07, 95% CI = −0.41, 0.27, P = 0.69).

There were no significant differences identified following meta-analysis for hip extension function when comparing high-intensity resistance to a control group (effect size = 0.23, 95% CI = −0.08, 0.54, P = 0.15; Figure 4). Insufficient studies were identified to perform a meta-analysis comparing high-intensity resistance to low-intensity resistance for hip extension.

The meta-analysis for knee extension function followed a similar pattern to hip abduction function and hip flexion strength (Figure 5). Although not significant, a larger effect size was identified when we compared a high-intensity resistance to a control group (effect size = 0.28, 95% CI = −0.12, 0.67, P = 0.17) than when we compared a high-intensity resistance and low-intensity resistance (effect size = −0.01, 95% CI = −0.35, 0.33, P = 0.97).

An individual study [39] (N = 208) reported a significant moderate effect (effect size = 0.38, 95% CI = 0.04, 0.71, P = 0.03) for hip adduction strength at ST follow-up time point favouring high-intensity resistance exercise when compared to control (Table 4). Another study not included in the meta-analysis [36] (N = 152) compared high-intensity and low-intensity resistance exercise and reported knee flexion strength, hip adduction, and hip internal and external rotation strength, which did not favour either group (Table 4).

There were no significant differences following meta-analysis for quadriceps femoris muscle size when comparing high-intensity resistance to low-intensity resistance (effect size = −0.07, 95% CI = −0.54, 0.40, P = 0.76; Figure 6). No studies were identified to perform a meta-analysis comparing high-intensity resistance to a control group for quadriceps femoris muscle size.

Only one study [29] (N = 39) reported on gluteal muscle volume (gluteus maximus: effect size = 0.22, 95% CI = −0.41, 0.85, P = 0.49, and gluteus medius: effect size = 0.10, 95% CI = −0.53, 0.72, P = 0.76) when comparing a high-intensity and low-intensity resistance exercise and identified no differences between groups.

### 3.3. Other Comparisons

There were insufficient studies that compared other exercise programs for a meta-analysis to be conducted for hip or knee muscle function or size, although results have been reported for individual studies.

A single study [36] (N = 152) comparing high-intensity resistance exercise with aerobic exercise resulted in a moderate effect (effect size = −0.51, 95% CI = −1.00, −0.03, P = 0.04) for hip internal rotation strength at the LT follow-up time point, favouring aerobic exercise. However, there was no significant effect favouring either exercise program for hip or knee function at any other follow-up time point (Table 4). The same study reported no benefits when comparing a low-intensity resistance exercise program to an aerobic exercise program for hip (abduction, adduction, flexion, and internal and external rotation) or knee (flexion and extension) function [36].

There were no benefits reported for quadriceps femoris muscle size in a single study [43] (N = 42) when comparing high-intensity resistance exercise to an aerobic exercise program at IT follow up (Table 4). The same study also reported no benefits for quadriceps femoris muscle size when comparing a low-intensity resistance program and an aerobic exercise program at IT follow-up (Table 4).

A single study [19] (N = 103) reported no benefits in a low-intensity resistance exercise program when compared to a control for hip (abduction, flexion, extension, and rotation) or knee (flexion and extension) function at ST follow-up (Table 4).

A single study [23] (N = 47) comparing low-intensity resistance exercise to multimodal reported no improvements in hip extension and abduction strength favouring either exercise program at ST follow-up (Table 4).

## 4. Discussion

The current review identified that a high-intensity resistance program likely increases hip abduction function slightly in people with hip OA when compared to a control group with low certainty of evidence. There were no identifiable differences in hip or knee function or muscle size when comparing a high-intensity resistance intervention with low-intensity resistance. Furthermore, a single study reported no benefits for hip or knee function following a low-intensity resistance program compared to a control. The limited benefits observed following high-intensity resistance exercise or other forms of exercise (e.g., low-intensity resistance exercise, multimodal) might be a consequence of the low number of studies identified in this review.

To produce significant improvements in muscle strength outcomes, an exercise program must be performed with sufficient intensity and duration to result in muscle adaptation, where initial strength gains result from neural adaptations [26] prior to muscle hypertrophy [44]. Intensity is commonly measured within resistance-based exercise programs as an overall percentage of one repetition maximum (% 1RM), with increases in strength generally being reported at higher relative training loads (e.g., >75% is classified as high intensity) [45]. Regardless of progression requirements and clinical condition, differences in dose–response relationships for trained and untrained participants suggest that the optimal mean training resistance is likely to be influenced by initial training status (80% and 60% of 1RM for trained and untrained individuals, respectively) [46]. Variability in other factors such as frequency, time, and type of exercise within the high-intensity resistance programs influence both strength changes and hypertrophy [22,44]. This review identified a consistent patten across a number of outcomes (i.e., hip abduction, hip flexion, and knee extension), whereby high-intensity resistance interventions produced larger effect sizes in improving hip and knee muscle function versus a control group when compared to the equivalent effect sizes associated with high-intensity resistance versus low-intensity resistance interventions. While this pattern supports the existence of a dose–response relationship, a general statement that high-intensity exercise results in greater improvement in muscle function when compared to low-intensity exercise in people with hip OA was not supported by the statistical analyses in this study. This might partly reflect the classification of one [29] of only two studies that compared high-intensity resistance and low-intensity resistance, where intensity of the intervention was based on velocity of movement. While a relationship between velocity and exercise intensity has been previously reported [30], it is not currently described within the ACSM guidelines.

A common change observed in people with hip OA is the decline of gluteal muscle volume (size) [6,11]. The meta-analysis in this review on quadriceps size, following implementing exercise interventions, found no benefit of high-intensity resistance versus low-intensity resistance exercise, consistent with muscle strength outcomes when comparing these two interventions. While previous research has identified an association between resistance-based exercise programs and muscle hypertrophy [21], we cannot confirm whether the changes in muscle strength identified in this review can be attributed to hypertrophy. Muscle size appears to be associated with objectively measured physical activity [47]; however, this relationship might depend on the overall intensity of the physical activity completed. Despite the fact that gluteal muscles are considered to be important stabiliser muscles of the hip [48,49], there was only one study identified in this review that reported on changes in hip muscle (gluteus maximus and gluteus medius) size. Future research should assess changes in both hip muscle size and strength following a rehabilitation program to identify if improved functional outcomes are associated with muscle hypertrophy in people with hip OA.

Adherence to an appropriately prescribed exercise-based rehabilitation program has previously been shown to positively impact hip and/or knee OA patient outcomes such as improvement in pain, general physical function [50,51], and overall quality of life [52]. A previous meta-analysis reported a minimum adherence of two resistance training sessions per week that would need to be completed to increase muscle strength [53]. While five of the six studies included within the current muscle function meta-analyses, reported good (>70%) [41] to high (>80%) [29,36,39,42] exercise adherence rates, one study [40] reported a low (53%) rate of exercise adherence. With low study numbers, this could have had a negative impact on some function outcomes (e.g., hip extension) included in this study. Future studies should not only aim to consider intermediate (13–24 weeks) or long-term (>24 weeks) intervention durations but also report on adherence rates and intervention fidelity to interpret the potential benefits in hip or knee muscle function in people with hip OA.

Overall, the methodological quality of all included studies was considered to have a low risk of bias. The overall certainty of the body of evidence ranged from very low to moderate for all outcomes. There was a low certainty of evidence and small important effect for high-intensity resistance interventions that likely increased hip abduction slightly when compared to a control, as studies were downgraded due to imprecision (small sample size) (Table A2) [39,41]. A smaller sample size might have the potential to negatively impact the internal and external validity of a study [54]. There was a moderate certainty of evidence for hip extension when comparing high-intensity resistance with a control group with studies downgraded due to imprecision (i.e., upper and lower confidence limit crosses an effect size of 0.5, which may reflect high variability between studies) [39,40,41]. Studies that reported on hip flexion [29,36,39,41], knee extension [29,36,40,41,42], and quadriceps femoris muscle size [29,43] when comparing high-intensity resistance with a control group or high-intensity with low intensity resistance were downgraded due to either imprecision (wide CI) or the risk of bias domain (e.g., small sample size or failure to blind outcome assessors), which could also be a potential limiting factor that can affect the internal validity of a study (Table A2, Table A3, Table A4 and Table A5) [55].

The strengths of this review include reporting on different types of exercise interventions across included studies for the meta-analyses (high-intensity resistance vs. low-intensity resistance and/or control groups) and the intervention classification (i.e., high-intensity, low-intensity, multimodal, and aerobic). Only studies that reported on an intervention duration of >6 weeks were included in this review, as strength improvements in shorter interventions (<6 weeks) may be due neuromuscular factors [26,27] and may not necessarily be a result of muscle hypertrophy. Multiple hip and knee outcome measures were considered in this review that included both muscle function and size.

Although the GRADE approach allowed a thorough evaluation of the quality of the body of evidence, the small number of available studies limited the outcome measures (i.e., hip abduction, hip flexion, hip and knee extension, and quadriceps femoris muscle size) and intervention duration (8–16 weeks) that could be included in a meta-analysis. A further limitation that should be considered when interpreting these findings is that the classification of exercise intensity for each intervention was based on the descriptions provided for each of included studies. For example, one study used similar resistance for comparison (exercise) groups, varying only in velocity of concentric and eccentric contractions, which may not have been enough difference in intensity for one group to impact hip or knee function outcomes compared to the other [29]. The optimal resistance training intensity is generally accepted at 80% 1RM for trained and 60% 1RM for untrained individuals [46]; however, this intensity (>60% of 1RM) may not have been maintained for the entire 8–16 weeks, due to varying levels or stages of intensity when progressing or regressing the participants throughout the intervention time.

## 5. Conclusions

This review identified that a high-intensity resistance intervention likely results in a slight increase in hip abduction function when compared to a control group for people with hip OA. However, high-intensity resistance may result in little to no difference in muscle strength or size when compared to low-intensity resistance. Additional high-quality studies are warranted to evaluate the influence of resistive intensity on hip and knee muscle function and size. The limited number of studies reporting on muscle size means that any changes in muscle function identified in this review cannot necessarily be attributed to muscle hypertrophy.

## Figures and Tables

**Figure 1 biology-10-01251-f001:**
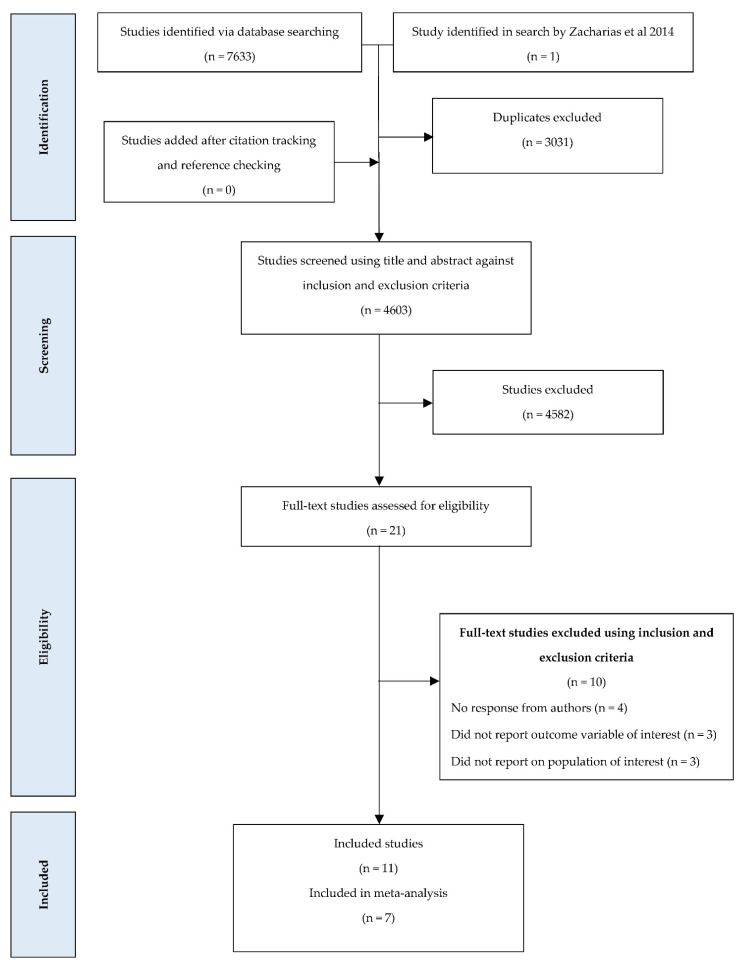
PRISMA flowchart summarising the yield of the keywords search and screening.

**Figure 2 biology-10-01251-f002:**
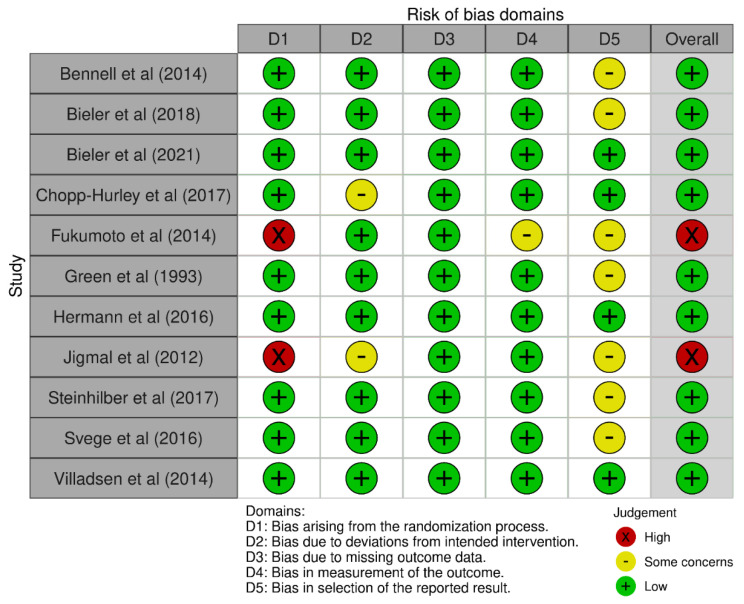
Risk of bias domain judgments across all included studies.

**Figure 3 biology-10-01251-f003:**
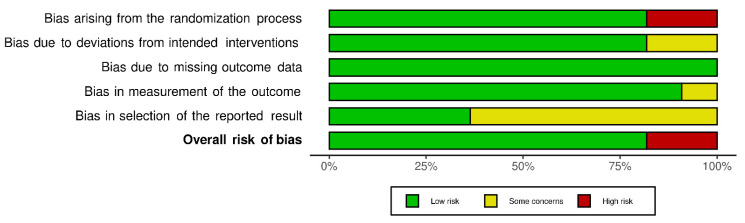
Summary of overall risk of bias across all included studies, presented as a percentage.

**Figure 4 biology-10-01251-f004:**
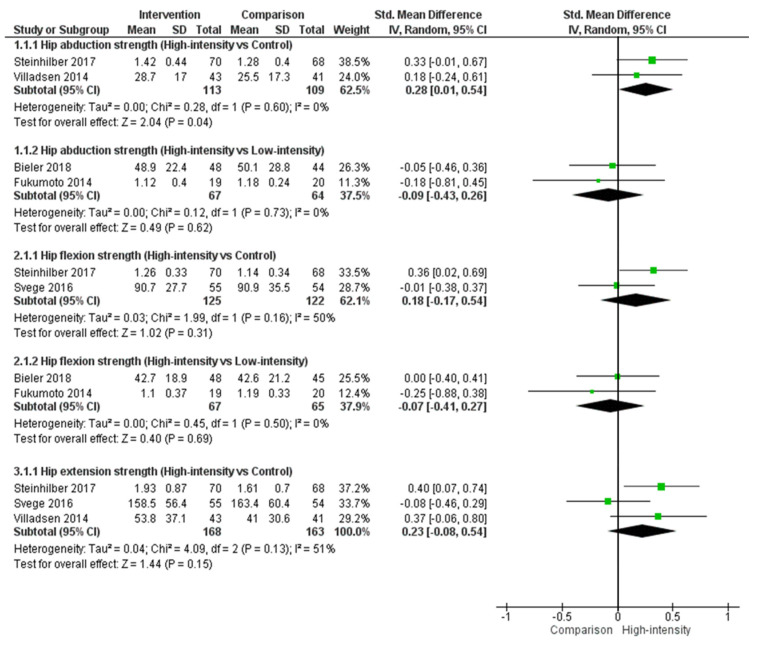
Hip muscle function when comparing a high-intensity exercise group to either a low-intensity exercise or control group in people with hip OA.

**Figure 5 biology-10-01251-f005:**
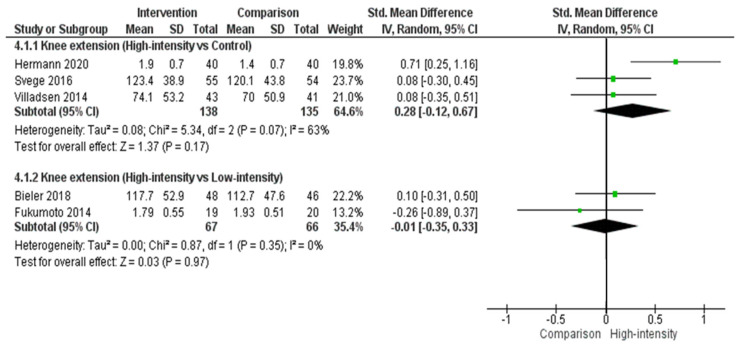
Knee extension function when comparing a high-intensity exercise group to either a low-intensity exercise or control group in people with hip OA.

**Figure 6 biology-10-01251-f006:**
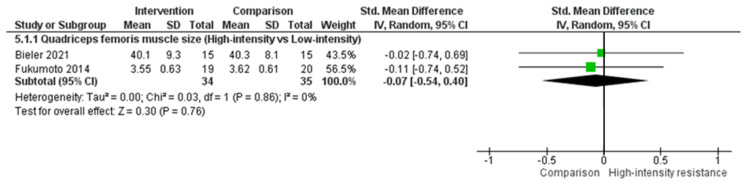
Quadriceps femoris muscle size when comparing a high-intensity exercise group to a low-intensity exercise group in people with hip OA.

**Table 1 biology-10-01251-t001:** Main concepts and keywords.

Concepts	Population	Intervention	Body Region
Keywords	[Osteoarthritis]	[Rehabilitation]	[Hip]
	OA	Physical therapy	
	[Arthritis]	Physiotherapy	
	Arth *	[Exercise]	

[ ], MeSH term; * root word/truncation search.

**Table 2 biology-10-01251-t002:** Inclusion criteria.

No.	Criteria
1	Must be in the English language
2	Must include human participants
3	Must include participants with hip OA (exclude juvenile idiopathic arthritis, femoroacetabular impingement, post hip arthroplasty, secondary OA, and dysplasia)
4	Must report on outcome variable of interest (i.e., muscle function or muscle size)
5	Must report on intervention of interest >6 weeks (rehabilitation/exercise program)
6	Must include original data of a peer-reviewed published paper (not conference proceedings, letters to the editor, or reviews)

**Table 3 biology-10-01251-t003:** Criteria used for classification of rehabilitation programs.

Rehabilitation Program	Classification Criteria
High-intensity resistance	Identified by resistance exercises involving a combination of body weight, externally weighted exercises, and elastic bands. The applied resistance during the main conditioning phase for higher intensity is ≥70% 1RM or multiple sets of <12 repetition range.
Low-intensity resistance	Identified by resistance exercises involving a combination of body weight, externally weighted exercises, and elastic bands. The applied resistance during the main conditioning phase for higher intensity is <70% 1RM or multiple sets of ≥12 repetition range.
Aerobic	Any other activity incorporating large body movements sustained for >10 min that aim to increase heartrate and oxygen uptake, excluding water-based exercises.
Multimodal	Exercise program that includes a combination of rehabilitation programs (e.g., hydrotherapy and low-intensity resistance).

**Table 4 biology-10-01251-t004:** Demographic, intervention, and outcome characteristics (muscle function data represented as SMD [95% CI]) within included studies.

Author	Participants	Intervention Duration	Comparison Groups (Classification Used in This Review)	Outcome Measures of Interest	Comparison; Effect Size SMD [95% CI] (Follow-Up Time Point; ST: 6–13, IT: 13–24, and LT: >24)		
Bennell et al. (2014) [19]	Hip OA: mixed unilateral and bilateral Group 1 N = 49 (26F, 23M) Group 2 N = 53 (36F, 17M) Mean age: Group 1: 64.5 ± 8.6 Group 2: 62.7 ± 6.4	12 weeks	Group 1—Low-intensity resistance exercise: Manual therapy (hip thrust manipulation, hip lumbar spine mobilisation, deep tissue massage, and muscle stretches), home exercises 4 times per week including HAbd strengthening (progressed through supine, standing, side lying, and standing wall press; 3 × 10 repetitions) and quadriceps strengthening (progressed through sitting elastic band press or KExt, partial squats, partial wall squats, sit-to stand and split sit to stand), balance and gait exercises Group 2—Control: Sham—No exercise instructions, inactive ultrasound	Hand-held dynamometer Isometric strength (Nm/kg): Unilateral hip strength (HAbd, HExt, HFlex, HIR, and HER) ^1^ Unilateral knee strength (KFlex and KExt) ^1^	Low-intensity resistance vs. control (positive—favours low-intensity resistance):		
					ST (week 13) ^2^	IT	LT
					HAbd: 0.13 [−0.28, 0.54] HExt: 0.32 [−0.09, 0.74] HFlex: −0.13 [−0.54, 0.28] HlR: −0.05 [−0.46, 0.36] HER: 0.28 [−0.13, 0.70] KExt: 0.07 [−0.34, 0.48] KFlex: 0.19 [−0.23, 0.60]		
Bieler et al. (2018) [36]	Hip OA: mixed unilateral and bilateral Group 1 N = 50 (34F, 16M) Group 2 N = 50 (33F, 17M) Group 3 N = 52 (36F, 16M) Mean age: Group 1: 69.6 ± 5.4 Group 2: 70.0 ± 6.3 Group 3: 69.3 ± 6.4	16 weeks	Group 1—High-intensity resistance exercise: Strength training—progressive in fitness centre with 3 mandatory resistance exercise machines; 1, leg press; 2, seated KExt; 3, standing HExt. Mixture of unilateral and bilateral exercises. 75% of 1RM for 10 repetitions × 4 sets Group 2—Aerobic exercise: ^2^ Nordic walking—12–14 on Borg scale (6–20) Group 3—Low-intensity resistance exercise: Unsupervised home-based exercises—hip ROM, stretching and strengthening exercises for the lower extremities using body weight and elastic bands for resistance	Good strength device (Ver 3.14) Isometric strength (Nm): Unilateral knee strength (KFlex and KExt) Hand-held dynamometer Isometric strength (Nm): Unilateral hip strength (HIR, HER, HFlex, HAbd, and HAdd)	High-intensity resistance vs. low-intensity resistance (positive—favours high-intensity resistance):		
					ST (week 8)	IT (week 16) ^3^	LT (week 52) ^3^
					KExt: 0.10 [−0.31, 0.50] KFlex: −0.04 [−0.45, 0.36] ^2^ HER: −0.05 [−0.45, 0.36] ^2^ HIR: −0.13 [−0.54, 0.28] ^2^ HFlex: 0.00 [−0.40, 0.41] HAbd: −0.05 [−0.46, 0.36] HAdd: −0.12 [−0.52, 0.29] ^2^	KExt: 0.11 [−0.30, 0.53] KFlex: −0.05 [−0.46, 0.37] HER: 0.00 [−0.41, 0.41] HIR: −0.15 [−0.57, 0.26] HFlex: −0.04 [−0.46, 0.37] HAbd: −0.04 [−0.45, 0.38] HAdd: −0.15 [−0.56, 0.27]	KExt: −0.01 [−0.47, 0.45] KFlex: −0.24 [−0.70, 0.22] HER: −0.29 [−0.75, 0.18] HIR: −0.33 [−0.80, 0.13] HFlex: −0.23 [−0.69, 0.23] HAbd: −0.19 [−0.65, 0.27] HAdd: −0.32 [−0.78, 0.15]
					High-intensity resistance vs. aerobic (positive—favours high-intensity resistance):		
					ST (week 8)	IT (week 16) ^3^	LT (week 52) ^3^
					KExt: 0.22 [−0.21, 0.65] KFlex: 0.10 [−0.33, 0.53] ^2^ HER: −0.05 [−0.48, 0.38] ^2^ HIR: −0.12 [−0.55, 0.31] ^2^ HFlex: −0.17 [−0.60, 0.26] HAbd: −0.16 [−0.59, 0.27] HAdd: −0.15 [−0.58, 0.28] ^2^	KExt: 0.10 [−0.34, 0.54] KFlex: −0.13 [−0.57, 0.31] HER: −0.21 [−0.65, 0.23] HIR: −0.32 [−0.76, 0.12] HFlex: −0.21 [−0.65, 0.22] HAbd: −0.32 [−0.77, 0.12] HAdd: −0.20 [−0.64, 0.26]	KExt: −0.02 [−0.49, 0.46] KFlex: −0.29 [−0.77, 0.19] HER: −0.34 [−0.82, 0.14] HIR: −0.51 [−1.00, −0.03] HFlex: −0.33 [−0.81, 0.15] HAbd: −0.36 [−0.85, 0.12] HAdd: −0.34 [−0.82, 0.14]
					Low-intensity resistance vs. aerobic (positive—favours low-intensity resistance):		
					ST (week 8) ^2^	IT (week 16) ^3^	LT (week 52) ^3^
					KExt: 0.13 [−0.31, 0.56] KFlex: 0.14 [−0.30, 0.57] HER: 0.00 [−0.43, 0.43] HIR: 0.01 [−0.42, 0.45] HFlex: −0.16 [−0.60, 0.27] HAbd: −0.10 [−0.53, 0.34] HAdd: −0.02 [−0.46, 0.42]	KExt: −0.02 [−0.47, 0.43] KFlex: −0.08 [−0.53, 0.37] HER: −0.18 [−0.63, 0.26] HIR: −0.14 [−0.59, 0.31] HFlex: −0.16 [−0.61, 0.29] HAbd: −0.25 [−0.70, 0.20] HAdd: −0.03 [−0.49, 0.42]	KExt: −0.00 [−0.50, 0.50] KFlex: −0.05 [−0.55, 0.45] HER: −0.04 [−0.54, 0.46] HIR: −0.15 [−0.65, 0.35] HFlex: −0.11 [−0.61, 0.39] HAbd: −0.12 [−0.62, 0.38] HAdd: 0.01 [−0.49, 0.51]
Bieler et al. (2021) [43]	Hip OA: mixed unilateral and bilateral Group 1 N = 15 (11F, 4M) Group 2 N = 12 (8F, 4M) Group 3 N = 15 (11F, 4M) Mean age: Group 1: 67.1 ± 3.9 Group 2: 69.1 ± 5.1 Group 3: 67.5 ± 5.2	16 weeks	Same participants as Bieler et al. (2018) [36] Only muscle size data used	MRI Cross sectional area (CSAcm^2^): Unilateral quads ^1^	High-intensity resistance vs. low-intensity resistance (positive—favours high-intensity resistance)		
					ST	IT (16 weeks)	LT
						Quads: −0.02 [−0.74, 0.69]	
					High-intensity resistance vs. aerobic (positive—favours high-intensity resistance):		
					ST	IT (16 weeks) ^2^	LT
						Quads: 0.06 [−0.70, 0.82]	
					Low-intensity resistance vs. aerobic (positive—favours low-intensity resistance):		
					ST	IT (16 weeks) ^2^	LT
						Quads: 0.09 [−0.67, 0.85]	
Chopp-Hurley et al. (2017) [37]	Hip and knee OA: mixed unilateral and bilateral Group 1 N = 12 (10F, 2M) Group 2 N = 12 (9F, 3M) Mean age: Group 1: 52.8 ± 6.4 Group 2: 54.9 ± 6.7	12 weeks	Group 1—High-intensity resistance exercise: Exercise classes within a sports and recreation facility incorporating static lower limb strengthening exercise, e.g., squats and lunges to elicit moderate activity in lower limb muscles, progressed over time Group 2—Control: No exercise	Fixed dynamometer Isometric strength (Nm/kg): Unilateral knee strength (KFlex and KExt) ^1^ Unilateral hip strength (HFlex and HExt) ^1^	ST (week 12) ^4^	IT	LT
					KExt: Group 1: 1.9 ± 0.5, Group 2: 1.6 ± 0.5 KFlex: Group 1: 0.7 ± 0.3, Group 2: 0.9 ± 0.3 HExt: Group 1: 1.4 ± 0.5, Group 2: 1.4 ± 0.7 HFlex: Group 1:0.9 ± 0.3, Group 2: 1.0 ± 0.3		
Fukumoto et al. ^5^ (2014) [29]	Hip OA: mixed unilateral and bilateral Total N = 46 (46F) Group 1 N = 19 Group 2 N = 20 Mean age: Group 1: 52.4 ± 9.2 Group 2: 52.5 ± 10.1	8 weeks	Both groups: daily home-based resistance training programs using elastic bands completing HAbd, HExt, HFlex, and KExt In addition, participants in the high-velocity group were instructed to perform the concentric phase of each repetition as rapidly as possible and then return through the eccentric phase in 3 s. Participants in the low-velocity group performed both the concentric and eccentric phases in 3 s Group 1—High-intensity resistance exercise: High velocity training—concentric phase of each movement as fast as possible and a slow eccentric phase total time = 3 s Group 2—Low-intensity resistance exercise: Low velocity training—completing concentric (3 s) and eccentric phases (3 s)	Hand-held dynamometer Isometric strength (Nm/kg): Unilateral hip strength (HExt, HFlex, and HAbd) ^1^ Unilateral knee strength (KExt) ^1^ B-mode ultrasound Muscle thickness (cm): Unilateral GMax, GMed, and Quads ^1^	High-intensity resistance vs. low-intensity resistance (positive—favours high-intensity resistance):		
					ST (week 8)	IT	LT
					HAbd: −0.18 [−0.81, 0.45] HExt: −0.24 [−0.87, 0.39] HFlex: −0.25 [−0.88, 0.38] KExt: −0.26 [−0.89, 0.37] GMax: 0.22 [−0.41, 0.85] ^6^ GMed: 0.10 [−0.53, 0.72] ^6^ Quads: −0.11 [−0.74, 0.52]		
Green et al. (1993) [23]	Hip OA: mixed unilateral and bilateral Group 1 N = 24 (18F, 6M) Group 2 N = 23 (17F, 6M) Mean age: Group 1: 65.7 Group 2: 68.0	6 weeks	Group 1—Multimodal exercise: Hydrotherapy with home-based exercises Group 2—Low-intensity resistance exercise: Home-based exercise made up of body weight and joint mobility exercises	Computerised dynamometer Isometric strength (N): Unilateral hip strength (HExt and HAbd) ^1^	Low-intensity resistance vs. multimodal (positive—favours low-intensity resistance):		
					ST (week 12) ^2^	IT	LT
					HExt: −0.29 [−0.86, 0.29] HAbd: −0.67 [−1.26, 0.08]		
Hermann et al. (2020) [42]	Hip OA Group 1 N = 40 (27F, 13M) Group 2 N = 40 (25F, 15M) Mean age: Group 1: 70.0 ± 7.7 Group 2: 70.8 ± 7.5	10 weeks	Group 1—High intensity resistance exercise: One hour supervised pre-operative progressive explosive unilateral resistance exercises including HExt, KExt, KFlex, and seated leg press. Exercise was performed explosively, with participants instructed to complete the concentric phase “as fast as possible” and the eccentric phase over “2–3 s” Group 2—Control Care as usual, no prescribed supervised exercise program	Nottingham power rig Isometric power (Watt/kg): Unilateral lower limb power (KExt)	High-intensity resistance vs. control (positive—favours high-intensity resistance):		
					ST (week 10)	IT	LT
					KExt 0.71 [0.25, 1.16]		
Jigami et al. (2012) [38]	Hip OA: mixed unilateral and bilateral—tested worse side (WS) and better side (BS) Group 1 N = 15F Group 2 N = 14F Mean age: Group 1: 60.8 ± 8.8 Group 2: 65.6 ± 7.8	(10 sessions for both groups) 10 weeks 20 weeks	Multimodal exercise: Land-based and aquatic exercises comprised of body weight exercises and stretching with aquatic muscle strengthening, whole body co-ordination and muscle relaxation Group 1: Fortnightly Group 2: Weekly	Hand-held dynamometer Peak force (kg): Unilateral hip peak force (HFlex, HExt and HAbd) Unilateral knee peak force (KFlex and KExt)	ST (week 10) ^2^	IT (week 20) ^2^	LT
					HFlex: WS: 20.1 ± 4.9, BS: 21.3 ± 5.8 HExt: WS: 19.9 ± 4.6, BS: 21.5 ± 5.4 HAbd: WS: 22.7 ± 3.6, BS: 22.2 ± 4.3 KFlex: WS: 13.3 ± 2.8, BS: 14.4 ± 2.3 KExt: WS: 29.4 ± 5.3, BS: 30.7 ± 6.0	HFlex: WS: 13.3 ± 3.5, BS: 15.2 ± 4.7 HExt: WS: 14.8 ± 5.2, BS: 17.1 ± 4.9 HAbd: WS: 15.5 ± 4.4, BS: 14.6 ± 4.8 KFlex: WS: 12.2 ± 6.7, BS: 12.9 ± 4.5 KExt: WS: 21.7 ± 8.2. BS: 22.7 ± 9.2	
Steinhilber et al. (2017) [39]	Hip OA: mixed unilateral and bilateral Group 1 N = 70 ^7^ Group 2 N = 68 ^7^ Group 3 N = 70 ^7^ Mean age: Group 1: 58 ± 19 Group 2: 60 ± 9 Group 3: 58 ± 10	12 weeks	Group 1—High-intensity resistance exercise: THu¨Ko exercise therapy—a progressive exercise program with a mixture of mobilisation, physical perception of movements, balance, and strengthening of hip muscles using basic exercise equipment, e.g., elastic bands, weight cuffs to a 15 on the Borg scale Group 2—Control: Non-treated control group Group 3—Placebo: ^2^ Ultrasound group—ultrasound machine invisibly turned off	Isomed 200 isokinetic dynamometer Isometric hip strength (Nm/kg): Bilateral hip strength (HAbd, HAdd, HFlex, and HExt) ^8^	High-intensity resistance vs. control (positive—favours high-intensity resistance):		
					ST (week 12)	IT	LT
					HAbd: 0.33 [−0.01, 0.67] HAdd: 0.38 [0.04, 0.71] ^2^ HFlex: 0.36 [0.02, 0.69] HExt: 0.40 [0.07, 0.74]		
Svege et al. (2016) [40]	Hip OA: mixed unilateral and bilateral Group 1 N = 55 (31F, 24M) Group 2 N = 54 (28F, 26M) Mean age: Group 1: 58.4 ± 10.0 Group 2: 57.2 ± 9.8	12 weeks	Group 1—High-intensity resistance exercise: Three exercise sessions x week of strengthening, functional and stretching exercises with 1 supervised by a physiotherapist—previously reported (70–80% 1RM) Group 2—Control: Patient education with list of exercises given to participants—3 sessions per week education with 8 week follow-up	Isokinetic dynamometer Isokinetic hip and knee strength (Nm): Unilateral hip strength (HFlex and HExt) Unilateral knee strength (KFlex and KExt) ^1^	High-intensity resistance vs. control (positive—favours high-intensity resistance):		
					ST	IT (week 16)	LT (week 40) ^3^
						KExt: 0.08 [−0.30, 0.45] KFlex: −0.07 [−0.45, 0.30] ^2^ HExt: −0.08 [−0.46, 0.29] HFlex: −0.01 [−0.38, 0.37]	KExt: −0.27 [−0.65, 0.11] KFlex: −0.27 [−0.64, 0.11] HExt: −0.37 [−0.75, 0.01] HFlex: −0.42 [−0.80, −0.04]
Villadsen et al. (2014) [41]	Hip and Knee OA (used only Hip OA data) Group 1 N = 43 (22F, 21M) Group 2 N = 41 (21F, 20M) Mean age: Group 1: 68.7 ± 8.4 Group 2: 68.6 ± 7.1	8 weeks	Group 1—High-intensity resistance exercise: NEMEX-TJR twice a week for 1 h supervised by physiotherapist and educational program Group 2—Control: Education program, written information provided	Muscle lab power, Ergo test. Unilateral lower limb muscle power (W): Unilateral KExt, HExt, and HAbd ^1^	High-intensity resistance vs. control (positive—favours high-intensity resistance):		
					ST (week 8)	IT	LT
					KExt: 0.08 [−0.35, 0.51] HExt: 0.37 [−0.06, 0.80] HAbd: 0.18 [−0.24, 0.61]		

Abbreviations: ST, short-term; IT, intermediate-term; LT, long-term; 1RM, 1-repetition maximum; ROM, rang of motion; Nm, Newton-metre; W, watts; F, female; M, male; kg, kilogram; sec, seconds; cm, centimetres; N, newtons; SE, standard error; SD, standard deviation; MVC, maximum voluntary contraction; HAbd, hip abduction; HAdd, hip adduction; HExt, hip extension; HFlex, hip flexion; HIR, hip internal rotation; HER, hip external rotation; KExt, knee extension; KFlex, knee flexion; GMax, gluteus maximus; GMed, gluteus medius; Quads, quadriceps femoris. ^1^ Most affected limb, ^2^ not included in meta-analysis (no other studies compared same interventions), ^3^ not included in meta-analysis (used earliest follow-up time point reported), ^4^ not included in meta-analysis (change scores reported), ^5^ comparison classification based on velocity, ^6^ not included in meta-analysis (only study to report muscle size outcome), ^7^ sex-specific breakdown not available per group, ^8^ mean of both limbs normalised to subject’s body weight.

## Data Availability

Not applicable.

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
