# Peer review of "Efficacy of Exercise-Based Rehabilitation Programs for Improving Muscle Function and Size in People with Hip Osteoarthritis: A Systematic Review with Meta-Analysis"

_biology, 2021, doi:10.3390/biology10121251_

Round 1
Reviewer 1 Report
Title: Efficacy of exercise-based rehabilitation programs for improv-2 ing muscle function and size in people with hip osteoarthritis: 3 a systematic review with meta-analysis
The article seems well built and brings evidence of a physiological phenomenon not yet fully understood and that certainly deserves further study.
Some points of revision, suggested to the authors to increase the level of the paper, are provided below
The abstract should be reinforced / I suggest including more details so the readers from the abstract will be stimulated to download the full text
I suggest following the SYSTEMATIC REVIEW AND META‐ANALYSIS: A PRIMER by Impellizzeri & Bizzini https://www.ncbi.nlm.nih.gov/pmc/articles/PMC3474302/
In fact, their suggest some details for META‐ANALYSIS:
- Data extraction must be accurate and unbiased and therefore, to reduce possible errors, it should be performed by at least two researchers. Standardized data extraction forms should be created, tested, and if necessary modified before implementation. The extraction forms should be designed taking into consideration the research question and the planned analyses. Information extracted can include general information (author, title, type of publication, country of origin, etc.), study characteristics (e.g. aims of the study, design, randomization techniques, etc.), participant characteristics (e.g. age, gender, etc.), intervention and setting, outcome data and results (e.g. statistical techniques, measurement tool, number of follow up, number of participants enrolled, allocated, and included in the analysis, results of the study such as odds ratio, risk ratio, mean difference and confidence intervals, etc.). Disagreements should be noted and resolved by discussing and reaching a consensus. If needed, a third researcher can be involved to resolve the disagreement.
Reviewer 2 Report
The authors conducted a systematic review and meta-analysis in a methodologically sound manner.
The topic is very interesting.
It is known that high-intensity resistance exercises can improve muscle strength in people with knee osteoarthritis; however, the effectiveness of these exercises in improving muscle function and size in people with hip osteoarthritis has received less attention.
Through this systematic review and meta-analysis, the authors found that high-intensity resistance programmes can slightly increase hip abduction strength, while no differences in muscle function or size were identified when comparing a high-intensity group with a low-intensity group.
I have no comments or suggestions to make to the authors. I only recommend to insert keywords other than those in the title. This allows you to optimise your search through search engines. Therefore, replace "hip; osteoarthritis; rehabilitation; exercise" with other keywords that are useful for searching your paper.
